# Local Chemotherapy of Skin Pre-Neoplastic Lesions and Malignancies from the Perspective of Current Pharmaceutics

**DOI:** 10.3390/pharmaceutics17081009

**Published:** 2025-08-01

**Authors:** Nadezhda Ivanova

**Affiliations:** Department of Pharmaceutical Technologies, Faculty of Pharmacy, Medical University of Varna, 9000 Varna, Bulgaria; nadejda.ivanova@mu-varna.bg

**Keywords:** nanomedicine, skin cancer, nano-carriers, topical drug delivery, dermal drug delivery, melanoma, squamous cell carcinoma, basal cell carcinoma, actinic keratosis, prodrugs

## Abstract

In the preceding and early stages of cancer progression, local drug delivery to pre-cancerous and cancerous skin lesions may be applied as an alternative or supplementary therapy. At present, 5-Fluorouracil, imiquimod, and tirbanibulin creams and ointments have established their place in practice, while several other active pharmaceutical ingredients (APIs) (e.g., calcipotriol, tretinoin, diclofenac) have been repurposed, used off-label, or are currently being investigated in mono- or combined chemotherapies of skin cancers. Apart from them, dozens to hundreds of therapeutics of natural and synthetic origin are proven to possess anti-tumor activity against melanoma, squamous cell carcinoma (SCC), and other skin cancer types in in vitro studies. Their clinical introduction is most often limited by low skin permeability, challenged targeted drug delivery, insufficient chemical stability, non-selective cytotoxicity, or insufficient safety data. A variety of prodrug and nanotechnological approaches, including vesicular systems, micro- and nanoemulsions, solid lipid nanoparticles, nanostructured lipid carriers, polymeric nanoparticles, and others, offer versatile solutions for overcoming the biophysical barrier function of the skin and the undesirable physicochemical nature of some drug molecules. This review aims to present the most significant aspects and latest achievements on the subject.

## 1. Introduction

Skin cancers are generally classified into melanoma and non-melanoma types. Altogether, they represent the most commonly encountered form of malignancy worldwide, and their incidence increases with the rise in the elderly population [1,2]. Melanoma is among the fastest progressing cancers overall; it accounts for approximately 1% of all skin cancer cases and is distinguished by a high mortality rate after metastasis [3,4,5]. Non-melanoma skin cancers (NMSCs), contrarily, are mostly slow-developing, locally invasive, and curable. The basal and squamous cell carcinomas (BCC and SCC, respectively) represent the prevailing 99% of all NMSCs, while the rest include some less common but aggressive skin malignancies, such as cutaneous lymphomas, Kaposi’s sarcoma, Merkel cell carcinoma, skin carcinosarcoma, and dermatofibrosarcoma [6,7,8].

Skin cancer etiology is considered to have a tight bond with the harmful impact of ultraviolet (UV) light from sun exposure. While the UV-C rays are short enough (λ = 200–290 nm) to be entirely absorbed by the ozone layer, the longer UV-B (λ = 290–320 nm) and UV-A (λ = 320–400 nm) rays manage to penetrate the troposphere and reach the human epidermal and dermal skin layers, respectively. Thereby, they induce cascade reactions of reactive species generation (e.g., reactive oxygen species—ROS; reactive nitrogen species—RNS), DNA oxidation and fragmentation, mutations, and subsequent cell damage [8,9]. In some cases, depending on the oxidative status and immunity of the body, these processes may lead to abnormal skin morphology such as keratinocytes deformation, hypertrophy (e.g., of stratum corneum or stratum spinossum), sebaceous hyperplasia, reduction in the Langerhans cells’ count and suppression of their function, dryness, hyperpigmentation, and not exclusively, benign or malignant neoplasia (skin tumors) [10,11,12]. Additional factors involved in skin cancer etiology are ionizing radiation, exposure to some chemicals (e.g., arsenic), viral infections (e.g., human papillomavirus—HPV), or other environmental triggers. Important pre-cancerous skin conditions to be timely addressed and treated are actinic keratosis (AK) and Bowen’s disease (BD), which manifest as dry scaly patches or red scaly patches, respectively, and hold the risk of progression into SCC [13,14] (Figure 1).

The cutaneous and transcutaneous dosage forms offer several advantages, among which are an easy and non-invasive application, avoidance of hepatic metabolism upon first pass and some systemic side effects, targeted therapy, possibility for controlled drug flux, and dose reduction. However, very few compounds are able to cross the SC, reach targets in the skin, and act selectively and safely without an explicit chemical modification, physical enhancement of diffusion, or technological formulation. There lies the difference between the numerous in vitro active anti-cancer drug molecules and the very few ones finding a practical application and reaching the clinic [15]. In this regard, the role of nanotechnologies for topical skin cancer prevention and treatment in the early or post-operative stages narrows down to ensuring the desired qualities of the therapeutic entity for dermal administration [16,17]. The efficient topical drug therapy of skin lesions requires a multi-target approach aiming at selective cytotoxicity, inflammation, and infection [18,19,20]. From a technological point of view, enhanced skin permeation, sustainable drug liberation, and stability are to be sought after. The various nanotechnologies may ensure or improve one or more aspects of the above [21,22,23,24,25]; the strong and weak sides of each particular nano-carrier will be reviewed, including the lipid-based nanoparticles and systems, the polymeric, and inorganic nanomaterials.

## 2. Skin Anatomy, Histology, and Physiology in Brief

The skin, being the largest organ in the human body, possesses a surface of about 2 m^2^ in adults and a thickness varying in the range of 0.5–4 mm. Well-known, inside out, it comprises the subcutaneous layer (also referred to as subcutis or hypodermis), the dermis, and the epidermis. The epidermis, as the outermost, non-vascularized layer of the skin, exerts a protective function against loss of water and electrolytes, physical and chemical stimuli, and entry of foreign bodies, including microorganisms. It consists of the superficial, non-vital, horny layer—stratum corneum (SC)—and the underlying vital epidermal tissue. The latter is additionally subdivided into four layers—the stratum basale, spinosum, granulosum, and lucidum, respectively [26,27,28,29]. The main cells in the epidermis are the keratinocytes. They undergo division in the basal layer, and one of two daughter cells migrates to the surface; during this process, the keratinocytes acquire a squamous morphology and finally differentiate into corneocytes—dead, anucleated cells filled with keratin. The tightly ordered corneocytes in a semi-fluid lipid matrix of ceramides, fatty acids, and cholesterol build the structure of SC, the nature of which will be further reviewed with respect to drug transport. Another major epidermal cell type is the melanocyte, responsible for the synthesis of melanin. The Langerhans cells and the Merkel cells in the epidermis are involved in the immune and sensory responses, respectively [26,27,28,29].

The primary function of the dermis, a connective tissue layer underlying the epidermis, is secretory and thermoregulatory. It also contributes to the skin’s defense mechanisms against pathogenic microorganisms by maintaining a slightly acidic pH of 5.4–5.9 on the surface as a result of the sweat and sebaceous glands’ secretion. The pH increases in the vital cutaneous layers up to 7.1–7.3 [30,31]. The hypodermis, a vascularized and neuro-supplied fatty tissue, serves as a fat depot and nourishes the upper skin layers. It regulates the body temperature through a neuro-reflective pathway in which the thermoregulatory center controlling the sweat gland secretion is involved [26,32].

## 3. Dermal and Transdermal Drug Transport

As for all environmental pests of chemical, biological, or physical origin, the main barrier for drug penetration into the deeper skin layers is SC. In between the corneocytes, the extracellular space in SC is organized in lamellar bilayers comprising cholesterol, fatty acids, ceramides, and structural proteins (filaggrin, loricrin, and involucrin). The structural integrity and the cohesiveness of the cellular and extracellular elements that build SC define its resilience and barrier function. A key role in this regard is attributed to the epidermal lamellar bodies situated in the stratum granulosum’s keratinocytes and responsible for the cornified layer’s lipid nourishment. Under standard physiological conditions, SC is partially or entirely impermeable to most substances, including drug molecules [33,34,35,36].

The most likely drug transport mechanisms across the SC are of the transepidermal type and involve bypassing diffusion through the extracellular lipid milieu (paracellular route) or transcellular passage. While the paracellular route is favored by drug molecules with moderate to pronounced lipophilicity (log P ≥ 2) and relatively low molecular weight (<500 Da), the trans-corneocyte pathway is an alternative transit for hydrophilic drugs whose nature does not allow sufficient solubility and diffusion through the lipid matrix. The latter transport mechanism requires the keratin hydration that regulates the permeability of corneocytes [37,38,39]. Supporting the dermal drug delivery are the transappendageal routes, which include diffusion through the hair follicles or the sweat glands [40]. Although considered a secondary mechanism of drug transportation, the transfollicular path has gained much attention in recent years. Particular reasons for that are the eloquent findings of increased drug uptake from hair-rich skin regions as compared to hairless skin areas, as well as the nanotechnological advancements that allow transfollicular drug targeting. The transfollicular drug transport is desirable when treating skin diseases affecting the pilosebaceous units (e.g., alopecia and acne) or when aiming at transcutaneous drug absorption [41,42,43,44].

Regardless of the transportation mechanism, three stages of drug entry through the skin layers could be distinguished. Penetration occurs when the drug has reached the SC and the vital epidermis (epidermal transport); permeation is defined as the transit of the drug to the vascularized layers (dermal transport); percutaneous absorption (transdermal; transcutaneous transport) is the process of the drug entering the bloodstream [45]. Each disease requires a drug delivery and preferential deposition targeted at specific skin structures. In the cases of pre-cancerous and cancerous skin lesions, such targets are the abnormal cell formations and the margins thereof. Up until stage II skin carcinomas, the abnormal tissue may spread from the epidermal layers to the dermis [29]. The affected skin areas are normally characterized by increased permeability [34].

## 4. Biopharmaceutical Aspects of Dermal and Transdermal Drug Delivery

### 4.1. Physiological Aspects

The structure and function of the skin dynamically change with age, as well as a result of environmental factors, care with cosmetic products, concomitant diseases, and intake of certain medications. Consequently, there are considerable intra- and interindividual variations in skin permeability. As the skin matures, its barrier function gradually improves—the water content reduces, the fat secretion enhances, and the immune defense develops. With aging, epidermal atrophy, reduced secretion of the sweat and sebaceous glands, and impaired superficial lipid mantle become notable; a decrease in the capillary resistance of the dermis and a loss of elasticity tend to occur [46,47]. These peculiarities of elderly skin, which is indeed the most affected by pre-cancerous and cancerous neoplasia, impose the use of predominantly lipid-based or amphiphilic drug vehicles or lipid-based nanotechnologies. In cases of dry skin (xerosis) or dry scaly skin patches (e.g., AK, BD, SCC), the same recommendations apply [48,49]. In oily skin regions or in cases of exuding or ulcerating skin lesions, primarily hydrophilic or amphiphilic vehicles and nano-carriers might be preferred for better results [50].

Although the water content in SC under standard conditions is low (10–15%), this layer is highly hygroscopic and can absorb water up to 50–75% of its own weight (after a bath or occlusive therapy, for example). The hydrated SC is characterized by increased permeability and a limited protective capacity. The underlying vital skin layers are substantially more hydrophilic, and their water content reaches up to 70% [51,52]. It should be acknowledged that the degree of skin hydration varies in a wide range depending on the air humidity, the outside temperature, the use of skin care products, and, not least, concomitant diseases and the intake of particular medicines [53,54,55]. Often, a pronounced skin dryness is observed in patients with diabetes, hypothyroidism, imbalanced sex hormones, or those receiving diuretics and retinoids. The administration of peripheral vasodilators and anticoagulants may potentiate drug absorption through the skin and, in cases of local dermal therapy, increase the risk of systemic side effects [56,57,58,59,60,61,62,63].

Skin metabolism is an additional defense mechanism against potentially hazardous molecules that have managed to bypass the SC. Isoforms of alcohol dehydrogenase, aldehyde dehydrogenase, flavin-dependent monooxygenases, carboxylesterases, and the cytochrome enzymes (CYP 2D6, 2E1, 3A4, 1A1, 1A2, 2C9) are expressed in the skin layers and/or the skin appendages. The enzymatic activity of the skin depends on the age (the activity of esterases, for example, increases as the body matures), the anatomical site (some enzymes are found only in specific skin areas), and some environmental factors (e.g., exposure to UV radiation). Cutaneous metabolism plays a key role in the prodrugs’ biotransformation into active metabolites. A classic example in this regard is corticosteroids [64].

### 4.2. Pathophysiological Aspects: Pre-Cancer- and Cancer-Related Impairment of the Skin’s Barrier Function

Several particularities of the pre-cancerous and cancerous skin lesions usually determine an impaired structural integrity and enhanced permeability compared to healthy skin. The pathologically increased vascularity is among the leading factors explaining the enhanced permeability and retention effect (EPR) inherent for most tumors and allowing a tumor-specific targeting of macromolecules and drug-carrying nanoparticles [65,66]. Other often encountered structural changes in the concerned skin regions include defective lamellar bilayers of the cornified layer’s extracellular lipid matrix, hyperplasia, hypergranulosis, hyperkeratosis, increased transepidermal water loss, and decreased hydration; a scaly appearance and increased pH on the surface are typical for most skin lesions [34].

### 4.3. Drug Aspects in Dermal Drug Delivery

Skin penetrability is predetermined by the drug’s chemical nature, and the most important characteristics related to it are the molecular weight, the hydrophilic–lipophilic balance, and the ionizability. The particle size of a substance when in a powder or suspension-type of dosage form and polymorphism, where applicable, stand out as factors as well. It is widely considered that passive diffusion through SC is possible for molecules below 500 Da, if not less, when other factors play a negative role. The octanol/water partition coefficient value (LogP_o/w_), on the other hand, being in direct relation with the practically significant distribution between dosage form and SC, should ideally fall in the range of 1–4 [67,68]. However, the lesser the polarity of a molecule and the greater the LogP_o/w_ value, the stronger the retention in SC and the more limited further drug diffusion to the vital skin layers is [69,70]. The presence of ionizable functional groups in the structure of the drug may hinder skin penetrability depending on the respective dissociation constant/s value/s (pKa). When charged, a molecule is more likely to step into intermolecular interactions, causing steric immobilization and preventing passive diffusion [71]. Ionization and the hydrophilic–lipophilic properties of an active pharmaceutical ingredient (API), as well as other undesirable physicochemical properties, could be modulated to a certain extent by obtaining prodrug derivatives [72,73,74]. Esterification or etherification reactions are frequently applied to engage hydroxyl (-OH/O-) and carboxyl (-COOH/COO-) functional groups with the purpose of achieving an enhanced permeation rate, reduced drug crystallinity, etc. [75,76]. A few examples of prodrugs for the topical treatment of skin pre-cancerous and pre-cancerous lesions are listed in Table 1.

### 4.4. Dermal Dosage Forms and Excipients

Dermal dosage forms could generally be classified into five categories, including liquid formulations (lotions, shampoos, foams), semi-solid forms (ointments, creams, gels, and others), powders for cutaneous application, irrigation solutions, and transdermal patches [89]. The last three, being either superficially active or designed for systemic drug uptake via percutaneous absorption, are not suitable formulations for targeting cancerous and pre-cancerous lesions. The topically active anti-cancer drugs rely mostly on liquid or semi-solid vehicles to ensure their stability during storage, retention on skin, liberation, and facilitated permeation [90,91]. Bases with amphiphilic properties, such as liquid emulsions, creams, bigels, nanobigels, emulgels, and foams, resemble the skin’s nature the best and are considered most appropriate for drug delivery to the vital epidermis and the dermis. Moreover, the utilization of lipid-containing multi-phase vehicles of the various types (oil-in-water, water-in-oil, or multiple emulsion bases) is in good agreement with the physiological needs of xerosis skin, inherent for most pre-cancerous or cancerous lesions. Exceptions include lesions accompanied by ulceration and open wound formation, for which the use of water-based formulations, such as sterile solutions and hydrogels, is recommended. In all cases, bases of low-to-moderate viscosity are preferable when one or more of the following is valid: a need for faster permeation and onset of action; application on larger and richly haired skin areas; application on massive, rough, uneven, and sensitive lesions [89,90,91]. Major drawbacks of the conventional dermal dosage forms in the context of skin cancer therapy are the limited potential for modified drug release, drug targeting, stabilization, and enhanced permeation rate [89,90,91]. Moreover, many of them suffer from low patient compliance and a high incidence of local adverse reactions [92]. Alternatively, dermal or microneedle-equipped intradermal patches (films) are being developed and studied as potential platforms for the topical delivery of APIs, including antineoplastic agents, for they allow prolonged retention and extended therapeutic effects [92,93]; however, in their classical variations, these dosage forms do not offer a solution for better skin tolerability or active drug targeting.

An important class of pharmaceutical excipients used in dermal dosage forms is the permeation enhancers. They represent a heterogeneous group of adjuvants united by their ability to facilitate the drug’s passive diffusion through SC [26,94,95]. Classic permeation enhancers are dimethylsulfoxide (DMSO) and other similar solvents—e.g., dimethylformamide, diethylene glycol monoethyl ether (ethoxydiglycol; Transcutol^®^), dimethyacetamide, and others; being very “good” solvents for a vast spectrum of molecules, they increase the skin’s permeability by dissolving and extracting lipids from the cornified layer and creating hydrophilic “channels”. DMSO, additionally, is known to denature keratin and other proteins in SC [96,97]. A disadvantage is the high active concentration of this permeation enhancer (>60%), which often causes skin irritation (erythema, skin dryness, urticaria, etc.) and limits its use [94]. Azone^®^ (1-dodecylazacycloheptan; laurocapram) is a strongly lipophilic compound (LogP_o/w_ 6.2) able to integrate into the phospholipid bilayer of corneocytes and increase the membranes’ fluidity; it is an effective absorption enhancer for both lipophilic and hydrophilic drugs and exerts its activity in relatively low concentrations (1–3%) [98,99]. Similar mechanisms of action are intrinsic for other permeation enhancers, such as oleic acid, pyrrolidones, terpenes, and terpenoids. The latter two main components of the essential oils may also increase the lipids’ solubility or cause vasodilation [100]. Other commonly used permeation enhancers are oxazolidinones, some keratolytics (urea, salicylic acid, etc.), and various humectants [101]. Except for chemical enhancers, physical processes and mechanistic approaches may also be applied to enhance absorption into and through the skin. Examples are ultrasound-assisted, iontophoresis-assisted, electroporation-assisted, and microneedle-assisted (trans)dermal drug delivery.

## 5. Nanotechnologies for Dermal Drug Delivery

Advancing technologies allow many of the desired qualities of a dermal formulation, namely, drug stability, prolonged retention on skin, extended drug release, enhanced permeation, targeted delivery, lowered toxicity, and improved tolerability, to be achieved by the use of nano-scaled drug delivery systems (Figure 2) [69,102]. Accordingly, the most recent reports on the local chemotherapy of skin pre-neoplastic lesions and malignancies involve nanotechnology-assisted approaches. In the context of dermal drug application, the suitable nano-carrier systems include vesicular carriers of lipid, polymeric, or hybrid nature, other lipid-based nanoparticles (e.g., solid lipid nanoparticles, nanostructured lipid carriers, nano- and microemulsions), dendrimers, and metal nanoparticles [5,103,104].

### 5.1. Vesicular Drug Carriers

The era of vesicular drug carriers began with liposomes in the 1960s. By being uni- or multi-lamellar vesicles comprising lipid bilayer(s) surrounding an aqueous milieu, they are able to accommodate lipophilic or hydrophilic APIs. Among their strengths are biocompatibility, biodegradability, and low immunogenicity. Following several generations of development, liposomal technologies still evolve and manage to overcome significant drawbacks from the past, such as the short biological half-life and the limited potential for targeted drug delivery. Thereby appear the cationic, stimuli-responsive, actively targeted, and long-lasting (stealth) liposomes [105]. Still, challenges to the large-scale and clinical introduction of liposomes remain: the long-term stability, the quality assurance of the manufacturing process for more complex (i.e., modified) formulations, the low drug loading and encapsulation capacity, and the production cost [106]. Meanwhile, different courses of modification have led to the establishment of other classes of vesicular systems with improved abilities in dermal and transdermal drug delivery. These include the ethosomes, transferosomes, invasomes, niosomes, polymersomes, pharmacosomes, phytosomes, cubosomes, glycerosomes, and chitosomes. Most of them share similar drug permeation enhancement mechanisms, involving one or more of the following: superficial skin deposition and drug release; diffusion of intact flexible vesicles through the skin layers and subsequent drug release; transfollicular targeting; vesicles’ fusion into cell membranes and subsequent cytoplasmic drug release; endocytosis by phagocytic cells [107,108]. In more detail, information about the various vesicular nano-carriers is revealed in Table 2.

### 5.2. Other Lipid-Based Nanotechnologies

Other lipid-based nanotechnologies emerging in the field of dermal and transdermal drug delivery, apart from the vesicular systems, are solid lipid nanoparticles (SLNs) and nanostructured lipid carriers (NLCs). Chronologically, SLNs appeared first in the 1990s as a result of Professor R.H. Müller and Professor M. Gasco’s attempt to overcome the drawbacks of liposomes and polymeric nano-carriers [108]. SNLs are core–shell organized nanoparticles composed of a solid (at room and body temperature) lipid nucleus and a surfactant crown. NLCs are a later development using liquid lipids in the particles’ core as well. The less organized (amorphous) and fluid state of the lipid matrix thereby obtained improves the drug loading capacity of the lipid nanoparticles. Both the SLNs and the NLCs are currently widely investigated in dermal drug delivery because of several key advantages they offer over liposomes, viz. improved drug stability (by SLNs), better physicochemical stability of the system (by SLNs and NLCs), additional occlusive-determined drug permeation enhancement mechanism through skin (by SLNs and NLCs), additional stratum corneum and cell membrane fluidization for improved skin penetrability (by NLCs), and greater potency in prolonged drug delivery (by SLNs and NLCs) [134,135,136,137].

The nano- and microemulsions, unlike classic emulsions, are also regarded as lipid-based nanotechnologies. While they cannot offer as many perspectives in drug release modification, active targeting, and drug stabilization as compared to the other members in the group, they still offer a few benefits as dermal delivery platforms. Both nano- and microemulsions do not require further formulation into a dosage form (e.g., creams, gels, lotions, foams, patches, etc.) as they serve as liquid dermal vehicles as well. The oil-in-water emulsions exert an additional hydration effect on the intracellular keratin and solubilize the superficial sebum, while the water-in-oil emulsions fluidize the extracellular lipid matrix of SC, which either way leads to an improved epidermal permeability [108,138,139]. Furthermore, nano- and microemulsions are obtained via comparatively simpler technologies and are distinguished by a narrow size distribution, highly increased active surface area, and uniform spreading on the skin [140,141]. Lastly, the microemulsions, which are formed spontaneously with significantly higher surfactant concentrations as compared to nanoemulsions, represent thermodynamically stable systems [142].

### 5.3. Polymeric Nanoparticles

Polymeric materials, such as natural or semi-synthetic polysaccharides, synthetic acrylates, and others, find broad application as structural units of dermal drug delivery dosage forms—e.g., semi-solid hydrogels and dermal and transdermal patches [143,144,145]. However, the majority of polymeric nanomaterials are usually not common in dermal or transdermal drug delivery because of their limited ability to permeate the skin layers [146,147]. Interesting representatives worth mentioning in the context of this review are dendrimers. These three-dimensionally branched macromolecules exert several unique properties that could be useful in targeted local cancer therapy. Particularly, they possess high surface functionality eligible for modification and the ability to reduce the toxicity of chemotherapeutic agents, to improve drug solubility, and to ensure controlled drug release [148,149,150]. Dendrimers are also known to improve the delivery and efficacy of photosensitizers used in PDT for cancer treatment and act as permeation enhancers [151,152]. Moreover, unlike the majority of polymeric nanotechnologies that suffer from low reproducibility, challenging scalability, and toxicity problems, dendrimers result in a highly monodisperse system with a precisely defined size range and are known for their good biocompatibility [153,154,155].

### 5.4. Inorganic Nanoparticles

Inorganic nanomaterials are a diverse group of nanoparticles that includes metal nanoparticles, mesoporous silica nanospheres, magnetic nanoparticles, quantum dots, graphene oxide nanosheets, and others [156,157]. Still, two things that unite the majority of members of this class are their intrinsic pharmacological activity and responsiveness to external stimulation (e.g., magnetic field, electric field, light, temperature, pH), with the latter being a valuable tool in active drug targeting and controlled release [158,159]. Because of all these, the applications of inorganic nanomaterials progressively extend from imaging and diagnostics to theranostics and gene and drug delivery [160,161]. However, it should be noted that the most substantial toxicity concerns are reported for the representatives of this class owing to decomposition, nuclear penetration, or impurities [155,162].

Metal nanoparticles, especially those comprising noble metals—silver, gold, palladium, and platinum—are of particular interest in skin pathologies. Although the relatively high toxicity of these materials often limits their use to external administration, they have been systemically reported to possess explicit antimicrobial and anti-tumor properties [163,164,165,166,167,168]. Furthermore, their large and highly reactive surface area allows functionalization with other therapeutic agents as well [168,169,170]. A common event in this respect is the utilization of silver and gold nanoparticles (AgNPs; AuNPs) in wound healing [171,172,173]. Some key features of AgNPs and AuNPs are the ability to generate ROS and improve the efficacy of PDT, as well as the ability to disrupt membranes and facilitate drug permeation and action [174,175,176]. On the other side, metal oxide-based nanomaterials, such as titanium dioxide (TiO_2_) and zinc oxide (ZnO), are an object of interest in sunscreens and other products for skin cancer prevention, although there have been serious safety concerns related to their use [177,178,179,180,181,182]. For example, the European Commission banned TiO_2_ (also known as E171) as a food additive in 2022 due to potential genotoxicity. While it is still allowed in non-food products, the search for safer alternatives is being prioritized [183].

## 6. Local Chemotherapy of Skin Pre-Neoplastic Lesions and Malignancies

### 6.1. 5-Fluorouracil (5-FU)

5-FU, an antimetabolite that stands in the way of thymidine synthesis and DNA replication, and one of the oldest broad-spectrum anti-cancer drugs, is considered a gold standard API in the local chemotherapy of skin pre-neoplastic lesions and cancers [85,184,185]. Topical 5-FU products, e.g., Carac^®^, Efudex^®^, and Fluoroplex^®^, range in 0.5% to 5% active concentration and are officially approved for the treatment of AK and superficial BCC [85,186,187]. However, numerous studies have revealed the efficacy of 5-FU against other skin cancer types as well, including SCC and melanoma [188,189,190,191,192,193,194,195]. Furthermore, data have become evident to support the mechanisms of action of the compound. For example, Peng et al. showed that 5-FU nanoparticles regulate the Wnt/β-catenin signaling pathway and thereby inhibit the proliferation of SCC cells and induce apoptosis [196]; Tian et al. found that 5-FU triggers anti-tumor immunity by activating the Stimulator of Interferon Genes (STING) pathway in cancer cells and enhancing the immune response against melanoma cancer cells [197].

5-FU is a BCS class III drug, and key disadvantages of the conventional topical therapy arise from its low skin penetrability. In this respect, 5-FU prodrugs, combinations with other therapeutics, and formulation into advanced drug delivery systems, including nano-scaled drug delivery systems, represent a tremendous interest today [198,199]. According to the PubMed database, only the count of clinical trials for the past 10 years involving 5-FU in skin cancer therapy exceeds 25 [200]. Apart from that, dozens to hundreds of scientific articles concern the subject annually. Table 3 summarizes the outcomes from some recent research studies on nanotechnology-assisted topical 5-FU delivery.

### 6.2. Imiquimod (IMQ)

IMQ is a topically active immunomodulator, particularly effective against superficial BCC. Imiquimod formulations for cutaneous use on the market (e.g., Aldara^TM^ 5% cream and Imiquimod 5% cream) are also indicated for the treatment of AK and genital warts. Lower active drug concentrations of 3.75% are recommended when the facial skin is to be treated [223,224]. IMQ’s primary mechanism of action is grounded on the interaction with Toll-like receptors 7 and 8, toward which the compound acts as an agonist. Thereby, it upregulates the inflammatory response of the immune cells (e.g., neutrophils, macrophages, dendritic cells) and induces apoptosis in tumor cells [223,224,225,226]. The so-far established pharmacodynamics of IMQ have encouraged scientists and physicians to suggest and prove the efficacy of the compound against other forms of skin cancer besides BCC; particularly, there have been promising results published on IMQ’s effectiveness in cases of nodular BCC and SCC and persistently positive margins of melanoma in situ [227,228,229,230,231].

Chemically, IMQ is characterized by a low-molecular-weight (240.3 g/mol) imidazoquinoline structure. Despite that, the API suffers an extremely low skin penetrability (<3%) [232]; this is likely due to its strict lipophilic nature, insufficient water solubility (reported to be between 0.6 and 7.5 µg/mL in the temperature range between 4 and 30 °C, respectively) [233], and unfavorably slow partitioning into the vital epidermis as a result of a high affinity to the commercially used lipophilic cream bases or the superficial SC upon application [234,235,236]. Furthermore, as a base (pKa 7.3), IMQ steps into an ionized state in an acidic environment, which may also contribute to its negligible permeation rate [237,238]. From the point of view of the safe local therapy of superficial BCC, the so-minimized risk of systemic uptake via the transdermal route and the subsequent occurrence of systemic adverse reactions might be beneficial [232]. However, it limits the therapeutic potential of the drug in nodular skin cancer formations and invasive skin tumors [235]. Table 4 presents a few recent attempts to improve the skin permeation rate and the local bioavailability of IMQ by using nanotechnological instruments.

### 6.3. Tirbanibulin

Tirbanibulin is a novel synthetic drug, recently approved (in 2020 by the FDA and in 2021 by the EMA) for the topical treatment of AK, particularly in the face and scalp area. It is available as 1% ointment under the brand name Klisyri^®^ [249]. Tirbanubulin has shown promising anti-tumor and anti-proliferative potential against skin cancer cell lines, including melanoma and SCC. Its primary mechanism of action is known to be microtubulin polymerization inhibition and cell cycle arrest in rapidly dividing cells. Additionally, studies have revealed interference of the API with the proto-oncogenic Src tyrosine kinase signaling pathway, p53 upregulation, and induced apoptosis via caspase-3 stimulation and poly (ADP-ribose) polymerase cleavage [249,250,251]. Clinical trials have concluded a relatively good tolerability of the drug and a lower occurrence of local adverse reactions in comparison to a control ointment. A key reason for the latter observations appears to be the substantially shortened period of treatment (up to 5 days) with tirbanibulin 1% in comparison to a two- to several-week duration of the treatment by other products with the same indication [249,250,251]. Currently, the API tirbanibulin and the product Klisyri^®^ are under patent protection.

### 6.4. Diclofenac

Diclofenac, a non-steroidal anti-inflammatory drug, is conventionally used to manage pain and inflammation in musculoskeletal disorders [252]. As a COX-2 inhibitor, the API blocks the synthesis of prostaglandin E2 (PGE2) (a hormone-like substance recognized as a promotor of tumor growth), which explains the established beneficial effects of the topical diclofenac therapy in the treatment of skin pre-cancerous lesions and tumors characterized by overexpression of the COX-2 enzyme and elevated levels of PGE2 [253,254]. The anti-tumor effects of the drug are also supported by data showing the activation of mitochondrial apoptosis pathways through the modulation of Bcl-2 protein expression [255]. To date, the topical application of diclofenac 3% has been proven to prevent the progression of AK into SCC and to reduce tumor burden in SCC and superficial BCC lesions [255,256]. Diclofenac 3% gel in 2.5% hyaluronic acid (under the trade name Solaraze^®^ 3% Gel) is FDA-approved for the treatment of AK [257]. Topical diclofenac monotherapy has shown limited efficacy in nodular and invasive skin tumors, whereas several combinations have been suggested for improved results—e.g., topical diclofenac and calcipotriol, topical diclofenac plus PTD, topical diclofenac plus CO_2_ laser ablation, and IMQ 5% [258,259,260]. A hybrid compound of diclofenac and hydroxytyrosol, a natural antioxidant and potential anti-cancer agent, synthesized by Tampucci et al., has shown enhanced cytotoxicity against SCC cell lines as compared to native diclofenac. Furthermore, introduced in the form of self-assembling nanomicelles, the newly discovered diclofenac ester demonstrated a reduced risk of systemic transcutaneous absorption and increased deposition in skin layers in comparison to Solaraze^®^ 3% gel [261]. While many other scientific reports on diclofenac nano-formulation exist, they do not specifically regard topical skin cancer treatment as a potential therapeutic indication.

### 6.5. Ingenol Mebutate (IM)

IM is a relatively new therapeutic agent that was approved by the FDA in 2012 for the topical treatment of AK. It emerged on the market as Picato^®^ 0.05% and 0.015% gel, with the lower concentration being indicated for sensitive skin zones (face and scalp). A key advantage of the local treatment with IM appeared to be the fast therapeutic response and the short duration of the therapeutic course (2–3 days) [262,263]. It is known that IM triggers protein kinase C- and neutrophil-mediated immune responses leading to rapid cell necrosis and cellular cytotoxicity [262,263,264,265]. Except for AK, IM has shown promise in the treatment of BD, BCC, and SCC [266,267,268,269,270,271]. However, recent reports have associated the use of IM with an increased risk of SCC in AK patients [272,273].

IM is a tetracyclic diterpene first isolated from *Euphorbia peplus* (milkweed; cancer weed; petty spurge)—a traditional herbal medicine, particularly used to treat pathological skin conditions [263,274]. Several successful attempts have been made for the chemical synthesis of IM and its further derivatization, aiming at a more efficient and scalable method for production and improved chemical stability with comparable pharmacological activity [275,276]. The compound possesses an eligible molecular weight for dermal delivery (430.5 Da) and overcomes the skin barrier via P-glycoprotein-mediated transport but is chemically unstable in alkaline media and is characterized by a very limited water solubility [262]. It is believed that the need for an organic vehicle (particularly isopropanol) for its formulation into a gel, as well as the low pH of the composition, maintained to improve stability, are among the leading causes of adverse skin reactions accompanying the topical use of IM [263]. By chemical modifications and investigation of the pharmacological properties and stability, Liang et al. have identified some key features for potent and more stable ingenol derivatives [277]. Bertelsen et al. have successfully synthesized a novel ingenol derivative, namely ingenol disoxate, which exhibits superior chemical stability, augmented potency, and comparable biological activity in relation to IM [278].

### 6.6. Vitamin D_3_ and Analogs

Vitamin D_3_ analogs (VD_3_A), such as calcipotriol (calcipotriene), calcitriol, tacalcitol, and maxacalcitol, are a first-line treatment for psoriasis because they are known to modulate the vitamin D receptor (VDR)-mediated responses, thereby promoting cell differentiation, inhibiting proliferation, enhancing apoptotic processes, and exerting immunomodulatory activity [279,280]. This mechanism of action has prompted physicians to suggest potential beneficial effects of VD_3_A on skin pre-cancerous and cancerous lesions, with the focus primarily directed towards calcipotriol. Over the last decade, several clinical trials have emerged to investigate the efficacy of combining calcipotriol (0.005% foam or ointment) with 5-FU (1–5% cream) as a local chemotherapy for AK and SCC [281,282,283,284]. In general, they have established improved efficacy and accelerated onset of action through the induction of anti-tumor T-cell immunity in comparison to 5-FU monotherapy [285]. In addition, calcipotriol has been recognized as an enhancer of the PDT of AK [286]. Although vitamin D_3_ and VD_3_A have shown independent anti-cancer properties overall, the so-far established effects seem to be selective and related to VDR expression in the tumor cell line [287,288,289,290]. Therewith, Podgorska et al. have explained the lack of eloquent anti-tumor activity against melanoma in vitro. However, the same authors emphasized the sensitizing effect of calcitriol or calcidiol (Vitamin D_3_ active metabolite and precursor, respectively) on melanoma cancer cells to ionizing (proton beam) radiation [291]. While many researchers have attempted to nano-formulate VD_3_A in order to achieve improved, prolonged, and/or safer antipsoriatic efficacy [292,293,294], no such attempts were found in the direction of improved therapy of skin cancers.

### 6.7. Retinoids

Synthetic retinoids evolve throughout several generations, seeking enhanced chemical stability, selectivity to retinoid receptor(s) subtypes, improved tolerability, and higher skin bioavailability at the expense of limited percutaneous absorption for the topically active representatives [295,296]. Generations I, III, and IV are primarily designated for the treatment of acne, while Generation II (etretinate and acitretin) is established for the oral therapy of psoriasis [295,296,297,298]. All-trans retinoic acid—tretinoin (first generation), adapalene (third generation), and tazarotene (third generation)—has shown promising results in skin cancer treatment and prevention [296,298]. Tretinoin has been available on the market the longest, first as a solution and a cream for dermal use (Retin-A^®^ 0.1%), and later as a gel, and as a microspheres-enriched modified-release gel (Retin-A micro^®^ 0.04–0.1%) [299]; therefore, it has been investigated in the direction of skin cancer the most and is even used off-label for prevention [299]. The compound acts as a non-selective agonist of the retinoic acid receptors (RARs; isomers α, β, and γ) and the retinoid X receptors (RXRs; isomers α, β, and γ) [295,296]. Independently, tretinoin is proven to downregulate pro-inflammatory nuclear transcription factors [295]. Altogether, the topical use of tretinoin is found to result in the induction of keratinocyte apoptosis, the enhancement of cell turnover, the inhibition of carcinogenesis, and the suppression of tumor growth [296]. Data are available for the effectiveness of topical tretinoin in the treatment of AK, BD, SCC, and BCC [299,300,301]. The selective RAR-β/RAR-γ agonist adapalene (available as Differin^®^ 0.1% cream or gel) is found to be efficient in reducing AK lesions and exerting anti-proliferative activity against melanoma cell lines [301,302,303,304]. Tazarotene (also a third-generation selective RAR-β/RAR-γ agonist; available as Tazorac^®^/Fabior^®^ 0.1% cream, gel, or foam) has additionally shown activity against BCC [305,306]; however, a clinical trial investigating its activity in patients with BCC nevus syndrome has disputed potency for this indication [307]. The purely RAR-γ selective trifarotene (fourth generation; available as Aklief^®^ 0.005% cream) is to date only suspected for its potential in skin cancers because of the mechanism of action [308]. Bexarotene, a third-generation RXR-selective agonist (referred to as a rexinoid representative), is available as Targetin^®^ 1% gel and is indicated for cutaneous T-cell lymphoma (CTCL) [296,309]. Another rexinoid, alitretinoin (as a 0.1% gel), is approved by the FDA for the topical treatment of Kaposi’s sarcoma [310].

Retinoids as topical therapeutic agents suffer several disadvantages, such as high potency of causing skin irritability and dryness, chemical instability (photosensitivity of the earlier generations), or extreme lipophilicity (for example, log P of bexarotene is 6.9) [295,296,309]. Thus, they represent ideal candidates for prodrug and nano-formulation. However, since skin cancers are not their primary indication, very few of the innovative drug developments with retinoids address advancement in this clinical direction. Sallam et al. accomplished enhanced activity against SCC and CTCL with an increased skin deposition rate when conjugating bexarotene with hyaluronic acid [311]. Warda et al. obtained several bexarotene analogs with increased potency to inhibit CTCL cells by nitrogen substitution/hydroxylation/ring contraction [312]. Shah et al. achieved considerably improved photostability and skin tolerability of tretinoin (as compared to Retin-A^®^ cream) when formulating it into SLNs and administering the drug-loaded particles in the form of a gel; however, the authors only suggest potential benefits in the topical therapy of skin cancers and pre-cancerous conditions [313].

### 6.8. Miscellaneous

Apart from the aforementioned APIs for the local chemotherapy of skin cancers and pre-cancerous lesions, which, one way or another, have reached clinical practice, there are numerous therapeutic entities in a research phase that are also showing promising results. These include repurposed antineoplastic agents, antibiotics, phytochemicals, inorganic nanoparticles, and others. The most recent developments rely on nanotechnology to maximize the efficiency and safety of the cutaneous application, leaning on all the arguments discussed in the Introductory Section of this paper. Also, an increasing number of researchers bet on dual or even triple combinations of nanotechnological approaches, an additional advanced drug delivery platform, and physical enhancers (e.g., iontophoresis, ultrasound, microneedles, etc.). Although the latter developments suggest much higher manufacturing and therapy costs, as well as a challenging reproducibility of the production technology, they hold the potential to solve more problems of the local chemotherapy of skin cancers [314]. Table 5 summarizes trends and achievements from the last few years.

## 7. Conclusions

Based on this review, local chemotherapy for skin pre-neoplastic and malignant lesions appears to be a rapidly evolving field. It is mainly driven by the high demand for non-invasive, more effective, targeted, safe, and patient-friendly therapeutic approaches. While conventional agents such as 5-FU, IMQ, tirbanibulin, IM, and diclofenac serve to date as clinical standards, their limitations (e.g., skin penetration, chemical stability, local tolerability, etc.) underscore the need for advanced delivery strategies. The various nanotechnological approaches offer enhanced skin permeation, improved drug stability, targeted delivery, and reduced systemic exposure. Numerous and diverse nano-scaled delivery platforms, including vesicular systems, SLNs, NLCs, polymeric, hybrid, and inorganic nano-carriers, have demonstrated promising preclinical and clinical advancements in the treatment of AK, SCC, BCC, and melanoma. Moreover, the repurposing of established APIs, along with the integration of natural compounds and prodrug strategies, expands the range of therapeutic options. For the sake of objectivity, challenges such as safety validation, large-scale manufacturing, and regulatory approval remain. As in many other therapeutic fields, the clinical translation of valuable laboratory findings requires a solid scientific background and further controlled clinical trials to support data for safety and efficacy.

## Figures and Tables

**Figure 1 pharmaceutics-17-01009-f001:**
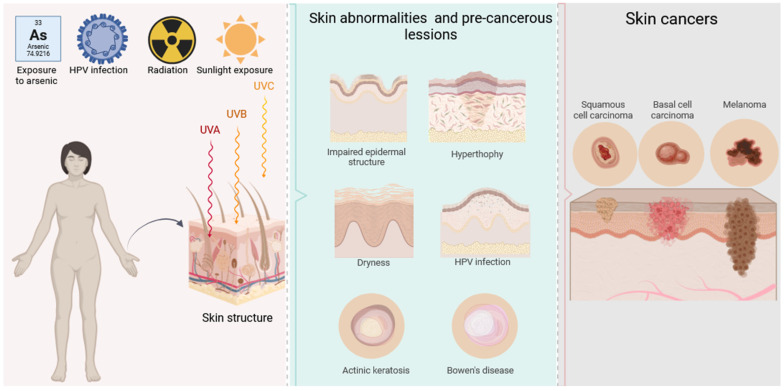
Skin pre-cancer and cancer etiology and types. Created in BioRender. Ivanova, N. (2025) https://BioRender.com/a97i800 (accessed on 27 February 2025).

**Figure 2 pharmaceutics-17-01009-f002:**
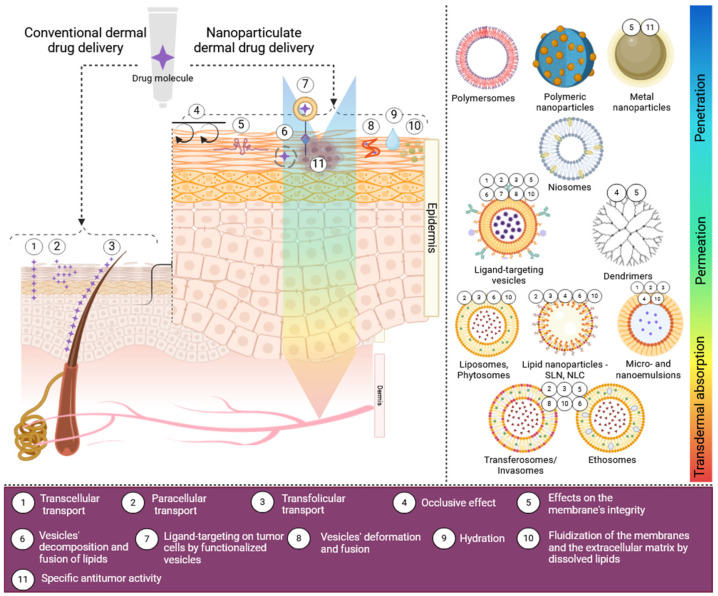
Conventional versus nanoparticulate dermal drug delivery. Created in BioRender. Ivanova, N. (2025) https://BioRender.com/m89a594 (accessed on 5 March 2025).

**Table 1 pharmaceutics-17-01009-t001:** Prodrugs for the topical treatment of skin cancerous or pre-cancerous lesions.

Drug	Prodrug(s)	Activity	Reference(s)
Temozolomide	Temozolomide hexyl ester	Melanoma (in vitro and in vivo animal studies)	[77]
Proto-Porphyrin IX	5-Aminolevulinic acid; Methyl-5-aminolevulinate; Butyl-5-aminolevulinate; Hexyl-5-aminolevulinate	In photodynamic therapy (PDT) of AK and BCC	[78,79,80,81,82]
5-Fluorouracil (5-FU)	1-Alkylaminocarbonyl 5-FU; N-Acyloxymethyl derivatives of 5-FU	AK BCC SCC	[83,84,85]
Nicotinic acid	Tetradecyl Nicotinate	Skin cancer prevention	[86]
Imiquimod	Imiquimod-oleic acid	AK BCC SCC Melanoma	[87,88]

**Table 2 pharmaceutics-17-01009-t002:** A comparative table of the various vesicular nano-carriers.

Type of Vesicle	Characteristics	Advantages Over Liposomes and Overall	Advantages in Dermal Drug Delivery	References
Liposomes	Flexible uni- or multi-lamellar vesicles composed of phospholipids, water, and cholesterol	-	Enhanced drug permeation through skin layers	[105,106,107,108,109,110]
Ethosomes and transethosomes	Flexible nano-sized vesicles composed of phospholipids, water, and ethanol in high concentrations *	Better compatibility with chemically unstable APIs	Enhanced drug permeation through skin layers; highly suitable for transdermal drug delivery	[109,110]
Transferosomes	Highly deformable nano-sized vesicles composed of phospholipids, water/water—alcoholic mixture, and an edge activator(s) (most often surfactants) **	High deformability	Enhanced drug permeation through skin layers; highly suitable for transdermal drug delivery	[111,112]
Niosomes	Nano-sized vesicles composed of nonionic surfactants, water, and cholesterol	Improved chemical stability; improved ability for drug stabilization; simpler preparation techniques and conditions; better reproducible technologies; lower production cost	Moderate enhancement of drug permeation through skin layers; suitable for topically active dermal formulations	[112,113,114]
Polymersomes	Nano-sized vesicles composed of amphiphilic block copolymers, surrounding an aqueous core	Increased loading capacity; increased chemical stability; improved ability for drug stabilization; suitable carriers for biomolecules such as peptides and proteins, incl. enzymes; adjustable release rate and suitability for controlled drug delivery	Generally, polymersomes are not suitable for dermal drug delivery due to their reduced penetrability; however, a useful application of these vesicles was recently discovered in the topical delivery of photoprotectors to the epidermis for skin cancer prevention [115]	[115,116,117,118]
Invasomes	Nano-sized vesicles similar to transferosomes but using terpenes as permeation enhancers	High deformability	Enhanced drug permeation through the skin layers	[119,120]
Pharmacosomes	Ultra-fine vesicular, micellar, or hexagonal aggregates carrying APIs that are covalently bound to the lipids	Higher entrapment efficiency; lowered risk of drug leakage	Improved control over drug liberation; reduced drug toxicity/adverse reactions	[121,122]
Phytosomes (herbosomes)	Nano-sized vesicles formed by the hydrogen binding of polyphenols to the hydrophilic choline heads of the liposomal phospholipids	Improved solubility, permeation, stability, and bioavailability of polyphenols	New perspectives and opportunities for the dermal drug delivery of polyphenols, incl. in skin cancer treatment	[123,124]
Cubosomes	Lyotropic nonlamellar liquid crystalline nano-sized vesicles usually comprise glyceryl monooleate, amphiphilic block copolymers, and/or other stabilizing agents	Higher entrapment efficiency; lowered risk of drug leakage	Bioadhesiveness; improved localized therapy of skin diseases; reduced drug toxicity/adverse reactions	[125,126]
Glycerosomes	Nano-sized vesicles composed of phospholipids, water, and glycerol **	Higher entrapment efficiency; longer shelf life	Enhanced drug permeation through the skin layers	[127,128]
Chitosomes	Chitosan-covered liposomes	Higher entrapment efficiency; improved stability	Bioadhesiveness; Enhanced drug permeation through the skin layers; Possibly own antibacterial, antifungal, and anti-tumor activity	[129,130,131,132,133]

* transethosomes contain a permeation enhancer in addition. ** other lipids, such as cholesterol, or additional surface-active or polymeric stabilizers, may also be used in the formulation.

**Table 3 pharmaceutics-17-01009-t003:** Nanotechnological advances in the topical delivery of 5-FU.

Formulation/Dosage Form	Nano-Carrier System	Therapeutic Combination	Result(s)	Reference(s)
Microemulsion	Microemulsion	n/a	Effectiveness in reducing carcinomatous areas in SCC-induced tissues; sustained drug release; significant skin permeation; minimal toxicity, enhanced patient compliance by reducing treatment duration and adverse reactions	[199]
Nanoemulsion	Nanoemulsion	n/a	Enhanced skin penetration and reduced irritation compared to conventional 5-FU 1% gel	[201]
Gel	Chitosan-functionalized nanoemulsion	n/a	Enhanced topical delivery of 5-FU; promoted retention of 5-FU in skin, and proven non-irritant properties of the nanoemulsion gel	[202]
Microneedle-assisted skin delivery	Liposomes	n/a	Enhanced 5-FU skin penetration and cytotoxicity against SCC; significant increase in 5-FU skin penetration by combining liposomes and microneedle technology	[203]
Liposomal suspension	Ultradeformable liposomes (transferosomes)	n/a	Enhanced 5-FU delivery to deeper skin layers	[204]
Liposomal suspension	Ultradeformable liposomes (transferosomes)	Resveratrol	Enhanced anti-tumor action of the drug combination of 5-FU and resveratrol against melanoma cells; improved skin permeation by ultradeformable liposomes	[205]
Liposomal suspension applied via subcutaneous injection or iontophoresis-assisted topical administration	Active-targeted immunoliposomes	Cetuximab	Reduced 5-FU permeation by iontophoresis as compared to injection and potential for reduced systemic side effects; increased 5-FU accumulation in viable epidermis; increased cellular uptake; effectively reduced proliferation of SCC cells	[206]
Liposomal suspension	Polymer-coated liposomes	n/a	Sustained release of 5-FU; increased cytotoxicity against epidermoid carcinoma cells as compared to uncoated liposomes	[207]
Niosomal suspension	Hyaluronic acid-coated niosomes	n/a	Improved targeting efficiency for 5-FU delivery, controlled drug release, and improved retention in skin	[208]
Microwave-assisted delivery of ethosomal suspension	Ethosomes	n/a	Enhanced skin penetration; increased cytotoxicity against human melanoma cells	[209]
Gel	Ethosomes	n/a	Enhanced local bioavailability of 5-FU; reduced skin irritation compared to marketed formulations	[210]
Gel	NLCs	Cannabidiol	Improved tumor remission and survival rate with reduced tumor volume in NMSCs; enhanced skin retention, excellent uptake, and deposition in skin layers	[211]
Gel	NLCs	n/a	Improved 5-FU permeation and skin retention; reduced skin irritation compared to plain 5-FU gel	[212]
Gel	SLNs	n/a	Enhanced permeability, sustained release, and cytotoxicity against melanoma and SCC; improved skin retention and permeation, and safety for topical applications	[213]
Gel	SLNs	n/a	Enhanced 5-FU skin penetration; reduced inflammatory reaction and reduced symptoms of angiogenesis in skin carcinoma-induced mice	[214]
SLNs suspension	SLNs	n/a	Improved cytotoxicity against melanoma cells compared to 5-FU solution; high entrapment efficiency	[215]
n/a	Chitosan nanoparticles	n/a	Enhanced encapsulation efficiency for 5-FU delivery; controlled drug release over 24 h; effectiveness against BCC	[216]
Nanofiber mat	Chitosan/Polyvinyl alcohol nanofibers	n/a	Effective treatment of BCC; high encapsulation efficiency and controlled release of 5-FU over 24 h	[217]
Nanofilms	Polyhydroxyethyl methacrylate/polyhydroxypropyl methacrylate/sodium deoxycholate nanoparticles	Iron–platinum nanoparticles	Controlled drug release with minimal side effects; anti-proliferative properties against BCC cells	[218]
Cream and gel	Gold nanoparticles	Gold nanoparticles	Reduced tumor volume in epidermoid carcinoma-bearing mice compared to free 5-FU, confirming enhanced anti-cancer efficacy and improved skin permeability	[219]
pH-responsive micellar hydrogel	Deoxycholic acid micelles	n/a	Improved anti-cancer activity against melanoma cells compared to 5-FU alone	[220]
Cream base	5-FU nanocrystals	n/a	Improved efficiency against epidermoid cancer cells as compared to the micro-sized crystals of 5-FU	[221]
Nanofibrous scaffolds	Nanofibers	Etoposide	Sustained drug release, significant cytotoxicity, and apoptosis against melanoma cells	[222]

n/a—not applicable.

**Table 4 pharmaceutics-17-01009-t004:** Nanotechnological advances in the topical delivery of IMQ.

Formulation/Dosage Form	Nano-Carrier System	Result(s)	Reference(s)
Patch	NLCs	Improved drug deposition in deeper skin layers as compared to commercial cream; patches enhanced patient compliance for topical skin cancer treatment	[234]
Microemulsion	Microemulsion	Limited ability of the microemulsions to improve the delivery of IMQ over Aldara™ cream	[235]
Gel	Microemulsion *	α-Tocopheryl polyethylene glycol 1000 succinate and oleic acid enhanced IMQ solubility significantly; the micellar formulation improved skin retention and delivery of the drug	[236]
Microemulsion	Microemulsion	IMQ 1% microemulsion delivered similar drug quantities to the epidermis as the commercial product with a 5-fold higher dose of 5%; lower risk of systemic absorption compared to the established product	[239]
Gel-like microemulsion	Microemulsion	Enhanced IMQ delivery to the skin and by a gel-like formulation with a suitable viscosity for topical application viscosity	[240]
Nanoemulsion	Nanoemulsion	Improved IMQ solubility and drug release; enhanced cytotoxicity against epidermoid carcinoma cell line compared to commercial formulation	[241]
Nanosuspension/nanoemulsion	Liposomes, nanocrystals, nanoemulsions, lipid nanocapsules	Best results with respect to permeation enhancement were achieved by using an IMQ nanocrystal suspension or nanoemulsion, which demonstrated a three- and five-fold increased drug delivery, respectively, in comparison to the commercially available cream	[242]
Liposomal suspension	Ultradeformable liposomes (transferosomes)	In vitro anti-melanoma activity	[243]
Nanostructured formulation (likely of micro- or nanoemulsion type) *	pH-responsive micelles	In vitro anti-melanoma activity; improved skin retention in comparison to Imunocare^®^ commercial product; pH-responsive drug release from micelles and possibility for selective drug release in tumor tissues	[244]
Ethosomal suspension	Ethosomes/transethosomes	Increased permeation rate of IMQ; improved retention and deposition into SC and the deeper epidermal and dermal layers as compared to Aldara™ cream; superior results were obtained with transethosomes (differing from the classic ethosomes by the presence of a permeation enhancer in their composition)	[245]
Nanosuspension	Polymeric nanoparticles (dextran nanocapsules)	Enhanced IMQ transdermal delivery, high encapsulation efficiency, and controlled drug release	[246]
Gel	β-Cyclodextrin-based nanosponges	Enhanced permeation and retention; sustained release of IMQ; greater inhibitory effect on fibroblast proliferation as compared to the pure drug	[247]
Nanosuspension *	Chitosan nanocapsules	Controlled IMQ release over 24 h; evaluated skin absorption; effective transdermal delivery	[248]

* not specified in the original paper but assumed from the given methodology.

**Table 5 pharmaceutics-17-01009-t005:** Novel therapeutic alternatives in the research phase for the topical treatment of skin pre-neoplastic lesions and malignancies.

Therapeutic Agent/Combination	Mechanism of Action	Formulation/Dosage Form	Nanocarrier System	Result(s)	Reference(s)
Dacarbazine	Alkylating chemotherapeutic agent (usually for parenteral administration)	Cream	NLCs	Superior anti-proliferative activity of the lipid nanoparticles with dacarbazine in comparison to pure drug against melanoma cell lines	[315]
		Gel	SLNs	Improved efficacy of the nano-formulation in the treatment of skin tumors induced in rats, as compared to free drug; stable formulation showing minimal side effects and potential in the topical treatment of melanoma	[316]
		Iontophoresis-assisted application	n/a	Iontophoresis appeared as a promising method to enhance the topical delivery of dacarbazine, potentially offering a safer alternative for melanoma treatment by avoiding the adverse effects of systemic administration; the study highlighted the necessity of stabilizing the drug in advance	[317]
		Gel	Nanosponges (polymeric)	Sustained drug release, effective permeation (73%), good biocompatibility, and superior inhibition of melanoma cell proliferation	[318]
Dacarbazine + tretionin	An alkylating chemotherapeutic agent combined with RARs/RAXs agonist	SLNs nanosuspension	SLNs	Effective inhibition of melanoma cell proliferation, induction of remarkable apoptosis, and inhibition of cell cycle progression and cell migration	[319]
		Nanosuspension	Transethosomes	Superior skin permeation and enhanced cytotoxic effects against cutaneous melanoma, indicating potential for effective topical treatment of skin cancer	[320]
Doxorubicin	Anti-tumor antibiotic, interfering with DNA and RNA synthesis (usually for parenteral administration)	Microneedle-assisted delivery	Hybrid cationic nanoparticles	Targeted accumulation in subcutaneous melanoma tumors in mice, induction of apoptosis, and suppression of tumor growth	[321]
		Gel	Oleic acid-grafted mesoporous silica nanoparticles	Improved permeation compared with the hydrogel containing free drug; improved doxorubicin’s cytotoxic effects toward epidermoid carcinoma cells	[322]
		Iontophoresis-assisted application	SLNs	Increased epidermal penetration; superior cytotoxic activity against SCC cells compared to pure drug solution; significant improvement in tumor tissue restriction in vivo	[323]
		Nanofibrous topical implantable delivery device	Core–shell implantable nanofibrous membranes	Controlled drug release, sufficient local concentration, allowing dose reduction and minimizing side effects; efficacy against melanoma tumors in vivo (in mice)	[324]
Doxorubicin + berberine	Anti-tumor antibiotic, interfering with DNA and RNA synthesis, combined with a naturally occurring isoquinoline alkaloid with proven anti-cancer activity and a complex mechanism of action	Gel	Mannose-conjugated NLCs	Improved permeation and skin deposition compared to conventional gel; potential for enhanced dual therapeutic approach for skin cancer amelioration based on in silico study	[325]
Doxorubicin + celecoxib	Anti-tumor antibiotic, interfering with DNA and RNA synthesis, combined with a non-steroidal anti-inflammatory drug, is effective in inhibiting skin cancer development	Microneedle-assisted delivery of liposomal gel	Liposomes	Enhanced in vivo anti-tumor efficacy of the combined liposomal gel against melanoma in comparison to single-drug liposomes; augmented skin penetration in the case of microneedle-assisted delivery	[326]
Bleomycin	Anti-tumor antibiotic, interfering with DNA synthesis (usually for parenteral administration)	Cream	Liposomes	Enhanced skin penetration by the liposomal formulation as compared to the free drug	[327]
Paclitaxel	Chemotherapeutic agents interfere with the normal function of microtubules during cell division	Anionic bicelles	Anionic bicelles	Effective penetration in SC; potential for treating skin cancer based on an in vivo study on mouse papillomas	[328]
Curcumin	A naturally occurring polyphenol with proven anti-proliferative and anti-tumor activity and a complex mechanism of action	Bioadhesive film (patch)	n/a	Enhanced drug penetration into the skin compared to a curcumin solution control; controlled drug release; effective against metastatic melanoma cells after topical application; superior tumor cell inhibition compared to a single dose of radiotherapy	[329]
Curcumin + anti-STAT3 siRNA	A naturally occurring polyphenol with proven anti-proliferative and anti-tumor activity, combined with gene therapy	Iontophoresis-assisted application	Liposomes	Inhibition of cancer cell growth in mouse melanoma cells, showing a statistically significant difference compared to treatments with either liposomal curcumin or STAT3 siRNA alone; iontophoretic administration demonstrated similar effectiveness in inhibiting tumor progression and STAT3 protein suppression compared with intra-tumoral administration	[330]
Resveratrol	A naturally occurring polyphenol with proven anti-proliferative and anti-tumor activity and a complex mechanism of action	Gel	Invasomes	High skin deposition and potency; high cellular uptake when tested on SCC; proven in vivo effectiveness in Ehrlich-induced mice models	[331]
Quercetin	A naturally occurring polyphenol with proven anti-proliferative and anti-tumor activity and a complex mechanism of action	Gel	Transferosomes	Enhanced skin permeation; lower cytotoxic concentrations against melanoma cells in comparison to quercetin conventional gel and solution	[332]
Diadzein + flaxseed oil	A naturally occurring polyphenol with proven anti-proliferative and anti-tumor activity, combined with a rich source of omega-3-polyunsaturated fatty acids source	Nanobigel	Nanobigel	Sustained release, improved drug permeation, and induction of apoptosis in epidermoid carcinoma cells	[333]
AgNPs + chlorhexidine	Silver nanoparticles with intrinsic anti-tumor activity, combined with a broad-spectrum antiseptic agent	Bioadhesive film (patch)	Green tea catechins-synthesized silver nanoparticles	Explicit and selective anti-melanoma activity of the nanosilver complex, fortified in the composition of the adhesive patch	[334]
AgNPs	Silver nanoparticles with intrinsic anti-tumor activity as a result of a complex mechanism of action	Nanosilver suspension	Myco-synthesized silver nanoparticles	Distinct anti-melanogenic activity	[335]
AgNPs + bixine	Silver nanoparticles with intrinsic anti-tumor activity, combined with a carotenoid with proven anti-proliferative and anti-tumor activity and a complex mechanism of action	Gel	*Bixa orellana* seed extract-synthesized silver nanoparticles	Moderate inhibitory activity against melanoma cancer cell lines; promising anti-cancer activity in vivo	[336]

n/a—not applicable.

## Data Availability

No new data were created or analyzed in this study. Data sharing is not applicable to this article.

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
