# Peer review of "Local Chemotherapy of Skin Pre-Neoplastic Lesions and Malignancies from the Perspective of Current Pharmaceutics"

_pharmaceutics, 2025, doi:10.3390/pharmaceutics17081009_

Round 1
Reviewer 1 Report
Comments and Suggestions for Authors
The manuscript offers a nice review of the field and summarizes current technologies and drugs for the treatment of pre-neoplastic lesions and malignancies. It also highlights future perspectives and challenges.
Overall, it is well written and structured.
Other comments by line:
Line 65: Unclear sentence – which fundamental prevents molecules to reach the practical application? I guess it is linked to previous sentence, which is also unclear – “narrows down to ensuring the opportunity for topical drug administration” – does this mean that the nanotechnologies should ensure topical delivery?
Line 241 and 266: For clarity reasons, please always use the same terminology – in line 241 “intradermal patches (films)” is used and in line 266 “microneedle-assisted (trans)dermal drug delivery” is used, although in both cases microneedle systems are (at least in part) the topic.
Line 245: there are also other chemical enhancers: e.g. Diethylene glycol monoethyl ether has been reported to enhance skin penetration for a variety of drugs, and is used in creams, lotions, microemulsions, aqueous gels and patches.
Figure 2: Not sure if the occlusion effect is specific to nanoparticulate drug delivery. For instance, the occlusion effect of Vaseline and its use is well known – and also evident from your previous introduction (line 165).
Line 378: In August 2022 EU has banned the use of TiO2 in food products. Maybe a reference to this and a case for TiO2 suitability for dermal application should be made here.
Line 453: No mention of diclofenac formulations in nanostructured systems – are there non described in the literature? You describe this topic nicely in line 521 for VD3A – something similar should be presented also for diclofenac (either examples in literature or a mention that there are none)
Line 480: use exact drug name not API for clarity reasons.
Line 493: wrong citation number
Line 557: idle is probably misspelled – correct should be ideal?
Comments on the Quality of English LanguageEnglish is sometimes not adequate from grammatic point of view. The manuscript should be revised by a native speaker for grammatical errors.
Some examples of sentences needing improvement:
The affected skin areas normally characterize with increased permeability [34].
Exception make the lesions accompanied with ulceration and open wound formation, by which the use of water-based formulations, such sterile solutions and hydrogels, is recommended.
Not by surprise, the most recent reports on local chemotherapy of skin pre-neoplastic lesions and malignancies involve nanotechnology-assisted approaches.
Being uni- or multi lamellar vesicles comprising of lipid bilayer(s) surrounding an aqueous milieu, they are able to accommodate lipophilic or hydrophilic APIs.
Author Response
Dear reviewer,
Thank you very much for your work on the manuscript and your contribution! The current document provides a point-by-point response to your remarks. The changes in the manuscript regarding your recommendations are highlighted in yellow.
Comment 1: Line 65: Unclear sentence – which fundamental prevents molecules to reach the practical application? I guess it is linked to previous sentence, which is also unclear – “narrows down to ensuring the opportunity for topical drug administration” – does this mean that the nanotechnologies should ensure topical delivery?
Response 1: A paraphrasing was done in this section for better clarity.
Comment 2: Line 241 and 266: For clarity reasons, please always use the same terminology – in line 241 “intradermal patches (films)” is used and in line 266 “microneedle-assisted (trans)dermal drug delivery” is used, although in both cases microneedle systems are (at least in part) the topic.
Response 2: A clarification in terminology was added in line 241.
Comment 3: Line 245: there are also other chemical enhancers: e.g. Diethylene glycol monoethyl ether has been reported to enhance skin penetration for a variety of drugs, and is used in creams, lotions, microemulsions, aqueous gels and patches.
Response 3: Thank you for adding this example; it was included in the text (in the section where the application of solvents as permeation enhancers is discussed).
Comment 4: Figure 2: Not sure if the occlusion effect is specific to nanoparticulate drug delivery. For instance, the occlusion effect of Vaseline and its use is well known – and also evident from your previous introduction (line 165).
Response 4: Figure 2 summarizes specific as well as non-specific but complementary mechanisms of drug transport and targeting offered by the various nano-carriers. The occlusive effect is typical for solid and semi-solid lipids regardless of the particulate state. Therefore, it is still evident by carriers that contain a substantial percentage of such materials – NLCs, SLNs, micro- and nanoemulsions, and others. It is also proven for some very small particles that manage to plug the skin pores.
Comment 5: Line 378: In August 2022 EU has banned the use of TiO2 in food products. Maybe a reference to this and a case for TiO2 suitability for dermal application should be made here.
Response 5: A comment regarding this information was included in the text along with a relevant reference.
Comment 6: Line 453: No mention of diclofenac formulations in nanostructured systems – are there non described in the literature? You describe this topic nicely in line 521 for VD3A – something similar should be presented also for diclofenac (either examples in literature or a mention that there are none)
Response 6: Thank you for this comment. Indeed, except for the given example of nanomicelles with a diclofenac derivative in the last rows of the paragraph, no other nanoformulations of diclofenac were found intended for topical skin cancer treatment or prevention. As you recommended, a comment on that issue was included.
Comment 7: Line 480: use exact drug name not API for clarity reasons.
Response 7: Done.
Comment 8: Line 493: wrong citation number
Response 8: Revised.
Comment 9: Line 557: idle is probably misspelled – correct should be ideal?
Response 9: Revised.
Comment 10: English is sometimes not adequate from grammatic point of view. The manuscript should be revised by a native speaker for grammatical errors.
Response 10: The grammar was corrected where pointed out by you. Furthermore, the whole manuscript was subjected to a thorough English editing.
Reviewer 2 Report
Comments and Suggestions for Authors
The manuscript is well-written; however, sections on skin anatomy, histology, and physiology &dermal and transdermal drug transport should be more summarized.
The drawbacks of conventional dermal dosage forms should be better highlighted.
The authors should mention a brief section on the merits of combining physical skin enhancers and nanocarriers, as it appears in Table 5.
Author Response
Dear reviewer,
Thank you very much for your work on the manuscript and your contribution! The current document provides a point-by-point response to your remarks. The changes in the manuscript regarding your recommendations are highlighted in blue.
Comment 1: The manuscript is well-written; however, sections on skin anatomy, histology, and physiology &dermal and transdermal drug transport should be more summarized.
Response 1: Some deletions and merging of sub-sections were performed to provide a more summarized view of these paragraphs. Information was retained where it had a direct relation to the following sections concerning the biopharmaceutical aspects of topical skin cancer therapy.
Comment 2: The drawbacks of conventional dermal dosage forms should be better highlighted.
Response 2: Done.
Comment 3: The authors should mention a brief section on the merits of combining physical skin enhancers and nanocarriers, as it appears in Table 5.
Response 3: A comment on that aspect was included before Table 5 with a relevant citation.